# Characterization of Sucrose-Impregnated Crystalline Glucose/Mannose Films as Moisturizing Wound Dressings and Their Significant Healing Effect on Deep Wounds in a Rat Model

**DOI:** 10.3390/bioengineering12040327

**Published:** 2025-03-21

**Authors:** Celine Chia Qi Wong, Kanako Tomura, Osamu Yamamoto

**Affiliations:** Graduate School of Science and Engineering, Yamagata University, 4-13-16 Jonan, Yonezawa 992-8510, Yamagata, Japan; celinewcq@gmail.com (C.C.Q.W.);

**Keywords:** wound healing, regenerative skin, sucrose, crystalline film, polysaccharide materials

## Abstract

Crystalline glucose/mannose film (G/M) demonstrated excellent water absorptivity and a high vapor transmission rate. The film had excellent wound healing performance. However, the crystallinity of the G/M tended to be disrupted when swollen, leading to a loss of mechanical strength. Herein, novel sucrose-impregnated G/M dressings were prepared to overcome this shortcoming. Sucrose has been claimed to promote angiogenesis and re-epithelization. This study evaluated the additive effect of sucrose on G/M films at different concentrations (G/M-0% (G/M-H_2_O), G/M-30% sucrose, and G/M-70% sucrose) regarding their material properties and wound healing performance. Both sucrose-impregnated G/M films demonstrated a lower degree of decrease in crystallinity than the control G/M-H_2_O film. This resulted in a significant improvement in stress and elongation at break for G/M-70% sucrose. In the in vivo study, G/M-70% sucrose was most effective in deep wound healing compared to other sucrose concentrations, with complete wound closure at 1 week. This was evidenced by the early regeneration of the mature epidermal layer, which promoted angiogenesis and the deposition of thicker and oriented collagen fibers. This study demonstrated the additive effect of sucrose on G/M, suggesting that the novel sucrose-impregnated G/M dressing is a promising candidate for deep wound healing.

## 1. Introduction

Wound healing is a complex process that aims to restore the damaged tissue after a skin laceration. In acute wounds, healing progresses smoothly through a series of healing phases accompanied by cellular activity, eventually regaining structural integrity. However, persistent inflammation during wound infection has been known to delay wound healing and lead to chronic wounds [1]. It has also been reported that the quality of life of chronic wound patients is greatly impacted due to physical limitations in carrying out daily activities and additional financial expenditures [2,3]. Therefore, wound dressing plays an important role in preventing wound infection, as it serves as a protective barrier to prevent the invasion of microorganisms. Over the past decades, modern wound dressing has been developed to provide wound healing under a moist environment, which prevents tissue dehydration, maintains cell vitality, and helps reduce pain [4]. Furthermore, a moist environment can promote cell migration and stimulate collagen synthesis, re-epithelialization, and angiogenesis [5,6]. In addition to providing a moist environment, dressings should possess other properties, such as the ability to absorb wound exudate, vapor permeability, non-adherence to the wound, and mechanical protection.

Commercially available hydrocolloid and polyurethane wound dressings have been clinically applied to epithelial damage but have a limited ability to absorb exudate from the wound, requiring short-term reapplication. This reapplication in a short period should be considered a high risk for bacterial infection from the outside environment. To achieve long-term application and short-term healing, therefore, various wound dressings have been studied with biopolymers such as silk fibroin, alginate, chitosan, gelatin, and glucose/mannose [7,8,9,10,11], which are favored over synthetic polymers as they are biocompatible, economical, and have chemical structures similar to extracellular matrix (ECM) [7,8]. Among biopolymers, sugars such as glucose have been reported to have moisturizing and wound-healing effects that are effective in moist environment treatment [9]. They prevent the accumulation of excessive fluid and expose the tissue to a microenvironment rich in proteinases and growth factors, while maintaining a moist condition at the wound site [4]. Additionally, the presence of large amounts of hydroxyl groups in the glucose/mannose structure can also act as active sites, allowing the incorporation of bioactive compounds into the structure, hence elevating the healing performance of the wound dressing material [10]. Moreover, reconstructive surgeons recommend that wound dressings have better mechanical properties than skin. This is because surgeons know that differences in elongation between skin and dressings cause friction and inflammation at the wound site. Wound dressings must withstand the stress of skin movement and daily activities without tearing. Therefore, the mechanical properties of wound dressings should be an important factor in wound healing.

On the other hand, granular sucrose was widely used in wound treatment in the early modern period, and its action was supposed to depend on water activity [11]. In other words, the effects of sucrose can be listed as follows: The low water activity in sucrose removes moisture from the wound site, resulting in a bactericidal effect. Sucrose is highly hygroscopic and forms hydrogen bonds with polar molecules, which reduces edematous inflammation, increases the water activity of the wound surface, and creates a moist environment for wound healing. In addition, sucrose is thought to produce a local osmotic effect, attracting macrophages and fibroblasts to the wound site and promoting granulation formation [11,12]. Additionally, the initial low oxygen pressure due to the presence of sucrose also causes macrophages to stimulate angiogenesis [12,13]. Nakao et al. [14] have clarified that sucrose accelerated the re-epithelization process via an up-regulating u-PA activator, which is crucial for keratinocyte migration. In developing countries, sucrose has been used for wound treatment due to limited access to modern wound dressings consisting of hydrocolloids and polyurethanes. A commercial ointment containing 70% (*w*/*w*) sucrose for superficial wounds, developed in 1964, has been marketed as a Class III drug in Japan. Based on the prescription documents, the active ingredient for wound healing is sucrose. However, this ointment is contraindicated for deep wounds, probably due to the ointment ingredients. This informative fact suggests that sucrose might be extremely effective in wound healing. As described above, numerous studies were conducted on the effects of glucose/mannose or sucrose on superficial wound healing, where they were used as a gel or viscous liquid.

In contrast, our previous report demonstrated that crystalline glucose/mannose films showed better healing ability in deep wounds than medical wound dressings, gels, and viscous liquid [15]. Crystalline glucose/mannose film possessed non-adhesion properties at the wound site, superior liquid absorptivity, and exclusive moisture retention properties. In order to achieve more rapid and greater wound healing than crystalline glucose/mannose film, the combination of crystalline glucose/mannose film and sucrose may be effective. However, there have been no reports on this combination. With the goal of further improving the healing efficiency of deep wounds, crystalline glucose/mannose films impregnated with sucrose were fabricated in the present work. The ability of crystalline glucose/mannose films impregnated with different concentrations of sucrose to heal deep wounds was assessed in animal experiments. The effects of sucrose concentration on the chemical structure and physical properties of crystalline glucose/mannose were also examined.

## 2. Materials and Method

### 2.1. Materials

In this study, glucose/mannose powder, sodium carbonate, and sucrose granules were used as received. Glucose/mannose powder and sodium carbonate were used to prepare the G/M crystalline film dressing. Glucose/mannose powder (PROPOL^®^ A) with a glucose/mannose molar ratio of approximately 1:10 was purchased from SHIMIZU Chemical Corporation (Hiroshima, Japan). Sodium carbonate was acquired from Nacalai Tesque Inc. (Kyoto, Japan). Sodium carbonate was obtained from Nacalai Tesque Corporation (Kyoto, Japan). Sucrose granules were used to prepare 30% (*w*/*w*) and 70% (*w*/*w*) sucrose solutions procured from Fujifilm Wako Pure Chemicals Corporation (Osaka, Japan).

### 2.2. Preparation of Glucose/Mannose (G/M) Film

Glucose/mannose (G/M) films were prepared by the casting method. A total of 1.0 g of glucose/mannose powder was completely dissolved in 100 mL of distilled water and stirred for 100 min at room temperature. Then, 50 mL of 0.01 mol/dm^3^ sodium carbonate solution was added to the G/M gel and stirred for 5 min. The G/M gel mixture was then poured onto a Petri dish and dried at room temperature. It should be noted that the G/M gel mixture in a Petri dish was sterilized daily with an ultraviolet sterilizer to prevent mold growth until a dry film was formed. The resulting dry G/M film was then rinsed with distilled water until a pH of 7 was reached. This process completely removed sodium carbonate from the G/M film. The resulting G/M film was autoclaved at 120 °C and then dried again at 60 °C in a sterilized container at atmospheric pressure.

### 2.3. Preparation of Sucrose Solution

Sucrose solutions of 30% (*w*/*w*) and 70% (*w*/*w*) were prepared by dissolving the respective amounts of sucrose granules in distilled water, as indicated in Table 1.

### 2.4. Preparation of Sucrose-Impregnated G/M Films

G/M films impregnated with 30% (*w*/*w*) and 70% (*w*/*w*) sucrose solutions were prepared by immersing the dried G/M films in their respective sucrose solutions for 30 min. For water impregnation without sucrose, the G/M film was immersed in distilled water for 30 min. The distilled water and sucrose aqueous solutions were sterilized by autoclaving prior to impregnation.

### 2.5. Characterization of G/M Film

G/M films in dry conditions, hydrated conditions in distilled water, and in 30% (*w*/*w*) and 70% (*w*/*w*) sucrose solutions were characterized for their crystalline structure and wound dressing properties. The G/M films under different conditions are described by the following abbreviations listed in Table 2.

#### 2.5.1. X-Ray Diffraction Measurement (XRD)

The crystalline structure of the G/M film under dry and hydrated conditions was analyzed using an XRD instrument (XRD Ultima-IV, Rigaku, Tokyo, Japan). The samples were measured by the parallel beam method under the condition of Cu Kα radiation (λ = 0.15418 nm) with a voltage and current of 40 kV and 40 mA, in the measuring range of (2θ) 3°–60°.

#### 2.5.2. Raman Spectroscopy Analysis

The types of bonds present in G/M and sucrose-containing G/M films were analyzed using a Raman spectrometer (NSR-3100Y, Japan Spectroscopy Co., Ltd., Tokyo, Japan). G/M-H_2_O was not characterized because the samples prepared in this analysis were dehydrated by drying for 24 h at room temperature after immersion in solutions prior to measurement. The experimental parameters were as follows: a laser wavelength of 532 nm (krypton ion laser) with an objective lens of ×20 magnification over a measuring range of 100 cm^−1^–4000 cm^−1^ at room temperature and atmospheric pressure.

#### 2.5.3. Mechanical Strength Analysis

Young’s modulus, the stress at break, and the elongation at break of the hydrated G/M films were determined using a universal tensile tester (Autograph AGS-J, Shimadzu Corporation, Kyoto, Japan). Ten film specimens per group were used for this mechanical evaluation. The film specimens used were rectangular in shape, with dimensions of 50 mm in length and 5 mm in width. The samples were first immersed in their respective solutions for 30 min, and then the average thickness of the films was obtained by measuring at three random points. Measurements were performed at a crosshead speed of 1.0 mm/min, with a sampling interval of 50 msec. The measurement was stopped when the film broke. Stress–strain curves were plotted, and the stress and strain at break can be determined directly from the graph. The Young’s modulus was then obtained from the slope of the curve and the maximum stress before rupture, according to Equation (1).(1)$$Young’s\;modulus\;\left(MPa\right)=slope\;of\;the\;graph=\frac{∆\;Stress}{∆\;Strain}$$

It should be noted that the dry mechanical properties of G/M were not evaluated in this study, as they were not used to evaluate wound healing in animal studies.

#### 2.5.4. Absorptivity Test

The G/M films used in the absorbance analysis were square in shape with 16 mm sides. The trimmed G/M film was weighed to obtain the initial mass of the dry film (*W_o_*). The dried film was then completely immersed in a phosphate-buffered solution (PBS) and kept in a 37 °C water bath. The weight of the swollen film (*W_t_*) at a given time was measured by first gently removing excess water on the surface of the film using filter paper. The measurement time interval was 15 min for the first 60 min, with a 1 h interval for the subsequent measurements for up to 12 h. The measurements were performed in triplicate. The absorption behavior of the G/M film was determined using Equation (2):(2)$$Water\;absorptivity\;\left(\%\right)=\frac{w_{t}-w_{o\;}}{w_{o\;}}\times 100$$
where *W_t_* is the weight of the expanded film at time *t* and *W_o_* is the initial weight of the dried film.

#### 2.5.5. Water Vapor Transmission Rate

The water vapor transmission rate (WVTR) of G/M hydrated in different solutions was determined in this study. Eight film specimens per group were used for this mechanical evaluation. Four groups were prepared including (i) blank control, (ii) G/M-H_2_O, (iii) G/M-30% sucrose and (iv) G/M-70% sucrose. G/M films in all groups except the blank control group were further covered with secondary dressing (Opsite Quick Guard; Smith & Nephew PLC, London, UK). The secondary dressing was used because it simulated the condition in which the secondary dressing was used to secure the G/M films to the wound during the animal study. The G/M films and secondary dressing were trimmed to a rounded shape and size to fit perfectly into the test tube. The test tubes were first completely filled with saline solution, then a doughnut-shaped Styrofoam ring, G/M film, and secondary dressing were placed. The weight of the test tubes was then measured and recorded as *W_o_*. The test tubes were incubated in a water bath at 37 °C and monitored for 14 days. The weight of the test tubes on day 14 was then measured and recorded as *W*_14_. Each group test was performed in triplicate. The WVTR of all groups was determined according to Equation (3):(3)$$WVTR\;\left(g/m^{2}/\mathrm{day}\right)=\frac{\left(w_{o}-w_{14}\right)}{A\times 14}$$
where *W_o_* and *W*_14_ are the initial weight and the weight at day 14 of the test tube, saline, G/M film, and secondary dressing, respectively. Also, A represents the area of the G/M film.

#### 2.5.6. Surface Roughness

The surface roughness of wound dressings may induce hyper-irritation at the wound site. To investigate the change in surface roughness caused by sucrose impregnation, the surface roughness of G/M films in all derivatives was determined by measurement using software (VK-Analyzer, Ver.2.4) installed on a confocal laser microscope (VK-9700 Violet Laser; Keyence Corporation, Osaka, Japan). The average surface roughness, *R_a_*, was obtained by measuring at ten random points of a sample to avoid any bias.

### 2.6. In Vivo Study of Full Thickness Wound

#### 2.6.1. Animal Operation

The G/M films were trimmed into squares with 16 mm sides and a thickness of 0.25 mm ± 0.02 mm under swollen conditions. The trimmed G/M films, sucrose solutions, and surgical instruments were sterilized using an autoclave (LBS-245; Tomy Seiko Co., Ltd., Tokyo, Japan) at 121 °C at 120 kPa for 20 min before surgery. The animal experiment was designed according to the guidelines approved by the Animal Care and Use Committee of Yamagata University (Figure 1). A total of eighteen male Sprague-Dawley rats weighing 250–300 g were used in this study to evaluate three groups of wound dressings. These included (i) G/M-H_2_O (control), (ii) G/M-30% sucrose, and (iii) G/M-70% sucrose. An inhalation anesthetic was first administered to sedate the rats, followed by an intramuscular injection of a triple anesthetic mixture. The amount of the triple anesthetic mixture was administered according to the rat’s body weight (kg): Medetomidine hydrochloride (NIPPON ZENYAKU KOGYO Co., Ltd., Fukushima, Japan), 0.15 mg/kg; Midazolam (Astellas Pharma Inc., Tokyo, Japan), 2.0 mg/kg; and Butorphanol tartrate (Meiji Seika Pharma Co., Tokyo, Japan), 2.5 mg/kg. Dorsal and ventral hair was then completely removed using a hair clipper and hair removal cream. The surgical site was then disinfected with 7% povidone-iodine and 70% ethanol to prevent contamination. Full-thickness skin incisions were then made on the abdomen using a 10 mm diameter trephine bar. Simultaneously, autoclaved G/M films were immersed in their respective solutions for 30 min prior to application to the wound site. The wounds were then covered with the hydrated G/M film of the respective treatment group and secured with a polyurethane secondary dressing. The wound healing status of all groups was evaluated at 1 and 2 weeks after treatment.

#### 2.6.2. Macroscopic Evaluation

The condition of the wounds, including the presence of granulation tissue and re-epithelialization, was monitored and photographed with a digital camera at weeks 0, 1, and 2. The wound area at week 0, 1, and 2 was determined by tracing the wound margin using Image J software (NIH, Bethesda, MD, USA; https://imagej.net/ij/). The residual wound area was determined using the following Equation (4):(4)$$Residual\;wound\;area\;\left(\%\right)=\frac{w_{t}}{w_{o}}\times 100$$
where *W_t_* and *W_o_* represent the wound area at 1 and 2 weeks and the wound area at week 0, respectively.

#### 2.6.3. Histological Evaluation

The rats for each group were sacrificed at 1 and 2 weeks after surgery for skin tissue collection. The regenerated tissue and surrounding healthy tissue were collected. The skin tissues were then fixed with 5% phosphate-buffered formaldehyde solution, followed by dehydration with an increasing concentration of ethanol solution, and finally embedded in paraffin blocks. Cross-sections of the skin tissues were then cut into 6 μm-thick sections for hematoxylin and eosin (HE), Masson’s trichrome (MT), and picrosirius red (PSR) staining (Muto Chemicals, Co., Ltd., Tokyo, Japan). Images of HE and MT staining were taken with a light microscope (BX53, OLYMPUS, Tokyo, Japan), and images of PSR staining were taken with a polarized light microscope (ECLIPSE Ci POL, Nikon, Tokyo, Japan).

#### 2.6.4. Angiogenesis Evaluation

The number of new blood vessels found in the regenerated tissue region was identified to evaluate the angiogenesis status and healing progress of the wound. Nine images of 2448 × 1980 pixels were taken at different positions of the regenerated tissue using a microscope. Blood vessels were identified by the presence of vascular endothelial cells and red blood cells. The image in pixels was then converted to area in mm^2^ using Image J, known as A, for further calculation. The number of new blood vessels per mm^2^ was determined using Equation (5):(5)$$Number\;of\;new\;blood\;vessels\;\left(\mathrm{units}/{\mathrm{mm}}^{2}\right)=\frac{N}{A}$$
where *N* is the number of new blood vessels found at specific pixels and *A* is the area of each image.

### 2.7. Statistical Analysis

All statistical data were reported as the mean ± standard deviation. After checking the statistical normal distribution, all results were analyzed by one-way analysis of variance (ANOVA), with Tukey’s post hoc test for multiple comparisons, using GraphPad Prism 10 software (GraphPad Software Inc., La Jolla, CA, USA). The results were considered statistically different at *p* < 0.05.

## 3. Results

### 3.1. Characterization of G/M Films in Different Conditions

#### 3.1.1. X-Ray Diffraction Analysis

For the XRD measurement, the G/M films impregnated with distilled water and sucrose aqueous solutions were prepared as described in Section 2.4, including the dried G/M film. Figure 2 shows the XRD diffraction peaks of G/M films in dry conditions, hydrated conditions in distilled water, and in 30% and 70% sucrose solutions. XRD diffraction peaks for the G/M films used were detected at 10°, 20°, and 22° in the diffraction angle. These peaks corresponded to the characteristic diffraction peak of the type II cellulose polymorph, which has an antiparallel β-sheet structure [16]. In addition, there was a peak shift to a lower diffraction angle when G/M films were in the hydrated state. This shift to lower angles corresponded to an increase in the interlayer distance in the β-sheet structure. Furthermore, it was observed that the intensity of the peaks decreased with the peak broadening that occurred when G/M films were in hydrated states. This is probably mainly due to the structural disorder caused by interlayer water and sucrose molecules interacting with G/M hydroxyl groups. However, G/M films hydrated in sucrose solutions showed a lesser degree of decrease in crystallinity. In addition, the degree of peak broadening was greater at lower sucrose concentrations. This can be explained by the stronger interaction between sucrose molecules and G/M chains compared to water molecules, which reduced the disordered crystal structure of G/M. The more sucrose molecules available, the stronger the intermolecular chain bonding. Therefore, sucrose molecules help to maintain a certain degree of crystallinity in the G/M structure in the hydrated state.

#### 3.1.2. Raman Spectroscopy Analysis

The functional groups present in the G/M, G/M-30% sucrose, and G/M-70% sucrose were determined after drying at room temperature for 24 h. The Raman spectra of the three groups (Figure 3) showed similar peaks at the proximal wavelengths of 3333 cm^−1^, 2606 cm^−1^, 1106 cm^−1^, 939.4 cm^−1^. These peaks were assigned to O-H stretching, C-H stretching, and the C-O-C glycosidic linkage of glucose/mannose structure [17,18,19]. In addition, the absorption peak at about 777 cm^−1^ corresponded to the anomeric region for D-glucose and D-mannose [19,20]. This characteristic absorption peak could be identified only in G/M and G/M-30% sucrose groups. This could be interpreted as the presence of a sucrose-rich deposit around the G/M structure when it was hydrated with 70% sucrose, thus masking the absorption peak. Furthermore, the absorption peaks at 1636 cm^−1^ and 1450 cm^−1^ could only be observed in G/M with sucrose groups. These peaks were assigned to H-O-H and the methyl-H-C-H bending of sucrose [19,20]. The intensity of these peaks also increased for the G/M-70% sucrose group. This shows that sucrose was successfully incorporated into G/M.

#### 3.1.3. Mechanical Strength Analysis

Young’s modulus, the stress at break, and the elongation at break of hydrated G/M in distilled water and sucrose solutions were also determined in this study. In Figure 4A,B, the Young’s modulus and stress at break of G/M increased with increasing sucrose concentration. The Young’s modulus and stress at break of G/M-70% sucrose were significantly improved compared to G/M-H_2_O, from 34.56 ± 4.08 MPa to 53.22 ± 2.54 MPa and from 4.63 ± 1.90 MPa to 26.14 ± 8.92 MPa, respectively. This indicates that immersion in a 70% sucrose solution improves the fracture resistance of the G/M film due to the presence of sucrose molecules. This fracture resistance was attributed to the presence of more hydrogen and van der Waals bonding through sucrose molecules, thus enhancing intermolecular interactions [21,22]. In addition, the elongation at break also increased with the increasing number of sucrose molecules present, from 34.79 ± 4.17% to 57.66 ± 13.09%. As mentioned in the XRD analysis, there was an increase in interlayer spacing due to the infiltration of water and sucrose molecules, thus improving the ductility of the G/M backbone [23]. From the above results, it could be concluded that the presence of sucrose improved the fracture resistance and ductility of the G/M film at the same time.

#### 3.1.4. Absorptivity Test

The ability to absorb excess fluid is one of the most important properties of a wound dressing. The absorption test was performed for 12 h at 37 °C to determine the absorption properties of G/M (Figure 5a). G/M achieved a maximum swelling ratio of 250 ± 10.32% after swelling in PBS for 2 h. During fluid absorption, the water molecules infiltrated into the G/M structure and interacted with the hydrophilic groups of the G/M structure. In addition, the ability of the G/M film to absorb water was said to be influenced by the availability of vacant hydroxyl groups in the G/M backbone [24]. Therefore, it was observed that the percentage of weight gain increased drastically at the beginning and leveled off when the G/M reached its equilibrium, with no more vacant hydroxyl groups available. Moreover, there was no weight loss and no change in state of the G/M film after soaking in PBS for 12 h. This indicates that G/M has is sufficiently able to absorb liquid while maintaining its original state (Figure 5b).

#### 3.1.5. Water Vapor Transmission Rate (WVTR) Analysis

The WVTR of hydrated G/M in distilled water and 30% and 70% sucrose solutions, coupled with a polyurethane secondary dressing, were evaluated and compared to a blank control (Figure 6). It has been reported that the optimal WVTR for commercially available wound dressings is in the range of 2000 to 2500 g/m^2^/day [25,26,27]. The WVTR for both the 30% and 70% sucrose-incorporated G/M dressing groups were found to be 2201.5 ± 83.1 g/m^2^/day and 2011.1 ± 46.2 g/m^2^/day, respectively, which were both within the optimal range. It was also found that the WVTR of the sucrose-impregnated G/M groups was significantly lower than that of G/M hydrated in distilled water, which was 3124.791 ± 143.3 g/m^2^/day. This indicates that the incorporation of sucrose molecules improves the moisture retention ability of the G/M film.

#### 3.1.6. Surface Roughness Analysis

The surface roughness of G/M films in dry and hydrated states are shown in Figure 7. The mean surface roughness, *R_a_*, of dry G/M was 4.12 ± 2.43 μm, which was significantly reduced when the G/M films were in hydrated states. The *R_a_* of G/M in water, with 30% sucrose, and with 70% sucrose, were 0.86 ± 0.31 μm, 0.72 ± 0.33 μm, and 1.07 ± 0.30 μm, respectively. This decrease in *R_a_* could be explained by the increased flexibility of the G/M film in the hydrated state. In addition, the reduced surface roughness of hydrated G/M makes it suitable as a wound dressing because it does not cause irritation or inflammation when applied to a wound [28].

### 3.2. In Vivo Wound Healing Study

#### 3.2.1. Macroscopic Evaluation

The wound healing statuses of the following dressing groups: G/M-H_2_O, G/M-30% sucrose, and G/M-70% sucrose were monitored at 1 and 2 weeks after surgery. Based on visual assessment (Figure 8a), the wound closure effect at 1 week was greatest in the G/M-70% sucrose group compared to all other groups. The residual wound areas of the G/M-H_2_O, G/M-30% sucrose, and G/M-70% sucrose groups were 29.98 ± 8.80%, 8.1 ± 5.68%, and 3.13 ± 4.75%, respectively. The percentage of residual wound area in Figure 8b showed that the presence of sucrose coupled with G/M film had significantly promoted wound closure at an early stage. At 1 week, granulation tissue was observed in G/M-H_2_O, indicating that epidermal regeneration had not occurred. In the G/M-30% sucrose group, pinkish tissue was found at the wound site. On the other hand, the G/M-70% sucrose treatment group achieved almost complete wound closure after 1 week of healing. After 2 weeks of healing, all treatment groups had achieved complete wound closure. However, the regenerated tissue condition was better in the G/M with sucrose treatment groups than in the G/M-H_2_O group. Because macroscopic evaluation only provided the superficial healing status of the wound, various tissue stains were performed to examine the healing progress of each wound in more detail.

#### 3.2.2. Histological Staining Evaluation

Figure 9 shows the overall healing status of the wounds and surrounding healthy tissue for each treatment group at 1 and 2 weeks after wound creation. The presence of epidermis for each treatment group at 1 week was further verified by HE staining (Figure 10a). No epithelial layer was found in the G/M-H_2_O group, whereas an epithelial layer was found in both sucrose-incorporated G/M groups. In addition, the wound of the G/M-70% sucrose treatment group had a thinner epidermal layer than that of the G/M-30% sucrose treatment group. The types of cells present at the wound site were also identified. An abundance of neutrophils was found in the G/M-H_2_O control group. On the other hand, macrophages and fibroblasts dominated the wound area of the G/M-30% sucrose and G/M-70% sucrose groups. In addition, more blood vessels were found in both sucrose treatment groups. This was further verified by angiogenesis assessment. Collagen deposition conditions were assessed by MT staining. The collagen fibers present in all treatment groups were fine fibers and in random orientations, corresponding to the early phase of collagen formation.

At 2 weeks of wound healing (Figure 10b), the epidermal layer was found in all three treatment groups. In addition, the epidermal layer thickness of the G/M-30% sucrose and G/M-70% sucrose groups further decreased, with the epithelial layer of the G/M-70% sucrose group more closely resembling the thickness of autologous skin epidermis. In the G/M-H_2_O control group, the number of neutrophils decreased along with the presence of macrophages and some fibroblasts. In contrast, the number of macrophages decreased in both sucrose treatment groups and was dominated by fibroblasts. MT staining showed that the orientation of the collagen fibers was improved in all groups. Coarse and dense collagen fibers were identified in the sucrose treatment groups. However, the collagen fibers of the control group were fine and sparse compared with those of the sucrose-treated groups. This was consistent with HE staining, which showed a smaller number of fibroblasts.

PSR staining (Figure 11) was used to differentiate between type III and type I collagen present at the wound site [29,30]. Type III collagen appears green to yellowish-green and type I collagen appears yellowish-orange to orange in PSR staining under polarized light microscopy [31]. At 1 week, green and yellowish-green collagen fibers were observed in all three treatment groups. This indicated that type III collagen dominated the wound site in all treatment groups, which was consistent with the MT staining results. At 2 weeks, denser yellowish-green collagen fibers were found in the G/M-H_2_O control group. In contrast, yellowish-orange type I collagen fibers were formed in both sucrose treatment groups. However, coarser and more oriented collagen fibers were observed in the G/M-70% sucrose group.

#### 3.2.3. Angiogenesis Evaluation

Angiogenesis at 1 and 2 weeks was evaluated by determining the number of blood vessels per mm^2^ of regenerated tissue (Figure 12). At 1 week, the number of blood vessels found showed that the G/M-70% sucrose group exhibited a significantly greater number of new blood vessels than the other treatment groups. This indicated that G/M-70% sucrose was effective in promoting angiogenesis. At 2 weeks after surgery, the number of blood vessels decreased in all treatment groups, indicating that wound healing progressed with healing time.

## 4. Discussion

Glucose/mannose (G/M) is a natural polysaccharide composed of β-1,4-linked D-mannose and D-glucose residues with a low degree of branching and acetyl groups randomly positioned at C-3 and C-6 of the glucose/mannose backbone, respectively [32,33]. Glucose/mannose in its natural state possesses a type I polymorph, which has parallel chain packing without intersheet hydrogen bonding [34]. Alkaline treatment is commonly used to change its crystal structure to the cellulose II polymorph, a more thermodynamically stable conformation in which intersheet hydrogen bonding exists [35]. This can be achieved by the addition of alkali, which removes the acetyl groups on the G/M backbone, resulting in a decrease in steric hindrance, thus leading to polymer chain association and a crystalline structure [36]. In this study, G/M samples in dry and hydrated states exhibited XRD patterns corresponding to the characteristic patterns of cellulose II. However, it was observed that the crystallinity of the hydrated G/M-H_2_O decreased significantly compared to the sucrose-impregnated G/M. This indicates that the sucrose molecules were able to fill in the gaps between the G/M polysaccharide chains, thereby enhancing the intermolecular chain interactions.

Properties such as mechanical strength, water absorbency, and water vapor transmission rate (WVTR) are some of the critical properties that affect the performance of a wound dressing. The mechanical properties of wound dressings are important because they need to withstand the stress of skin movement and daily activities without tearing, while allowing for ease of movement [24]. In this study, water and sucrose molecules penetrated the crystalline G/M structure and induced a certain plasticizing effect on the G/M film, as evidenced by the increased flexibility of the films in the hydrated state. It was reported that the plasticized polymer with a more compact microstructure exhibited higher fracture toughness and elongation [37]. The introduction of sucrose molecules into the structure of the G/M with 70% sucrose solution had significantly improved the toughness and elongation of the G/M film. It was assumed that the G/M-70% sucrose had a better ability to maintain the crystallinity of the G/M structure by strengthening the intermolecular sheet attraction through stronger hydrogen bonding and van der Waals interactions with a greater number of sucrose molecules. The XRD results for G/M-70% sucrose supported this consideration. In addition, a wound dressing should also have a high fluid absorption capacity to remove excess wound exudate from the wound site to prevent wound infection, which may lead to further complications. At the same time, the dressing should allow an optimal amount of water vapor to escape from the wound site. A wound dressing with a relatively high WVTR will result in the dehydration of the wound surface, while a low WVTR results in an accumulation of fluid at the wound edge, leading to wound maceration [38]. Therefore, a wound dressing with excellent water absorption and optimal WVTR is required to provide a moist wound healing environment for effective wound healing. In the absorbency test, G/M showed excellent water absorbency because it is highly hydrophilic due to the presence of a large number of hydroxyl groups in the structure. In addition, sucrose-impregnated G/M showed optimal WVTR and surface roughness within the recommended range. It was confirmed that the incorporation of sucrose molecules into the crystal structure of G/M can improve the WVTR of hydrated G/M films. This reduction in WVTR is largely due to the tight packing of the layered crystal structure of G/M in the presence of sucrose molecules [39]. Based on the above characterization, G/M-H_2_O, G/M-30% sucrose, and G/M-70% sucrose showed ideal properties as wound dressings, with the sucrose-impregnated G/M films showing optimal WVTR and better mechanical performance. Therefore, these G/M films in the hydrated state were then evaluated for their wound healing performance though an in vivo study.

Wound healing is a complex process involving four overlapping phases, including hemostasis, inflammation, proliferation, and remodeling. Hemostasis occurs immediately after trauma when platelet aggregation occurs at the wound site to inhibit bleeding. Once bleeding has stopped, neutrophils are the first to arrive at the wound site to clean the wound. Monocytes then differentiate into macrophages to phagocytose bacteria, debris, and neutrophils. The proliferative phase begins when M2 macrophages recruit fibroblasts, endothelial, and epithelial cells to the wound site for extracellular matrix deposition, angiogenesis, and epidermal regeneration [40]. Remodeling involves activities such as collagen maturation (type III to type I collagen), vascular maturation, and regression [40].

At 1 week of in vivo healing, the G/M-30% sucrose and G/M-70% sucrose treatment groups had significantly less residual wound area than the G/M-H_2_O control group, and re-epithelialization had occurred in both sucrose groups. According to Q. Li et al. [41], the thickness of the neo-epidermis changes over time, with the thickness decreasing as the epidermal layer matures with healing time. A thicker neo-epidermis was observed in the G/M-30% sucrose group. This indicated that the epithelium formed was still immature. Actually, pinkish regenerated tissue was found at the wound site. In contrast, the regenerated epidermal layer of G/M-70% sucrose more closely resembled autologous tissue. It was also found that macrophages and fibroblasts dominated the wound site of both sucrose-containing G/M dressings, whereas large numbers of neutrophils were observed in the control group. The formation of new blood vessels, known as angiogenesis, in regenerated tissue is important because it is responsible for transporting nutrients and oxygen to the cells to carry out wound healing activities.

At 1 week, the number of new blood vessels increased significantly with the higher concentration of sucrose incorporated in the G/M film. These results indicated that the healing phase for the G/M-H_2_O treatment group was in the early inflammation phase, the G/M-30% sucrose group was in the middle of inflammation and proliferation, and the G/M-70% sucrose group was in the proliferation phase. Therefore, these results meant that sucrose has an additive effect on G/M film by promoting re-epithelialization and the formation of new blood vessels in the early phase of wound healing. At 2 weeks after surgery, the presence of macrophages and fibroblasts along with sparse and fine type III collagen fibers in the control group indicated that wound healing was in the proliferation phase. In the G/M-30% sucrose and G/M-70% sucrose groups, the wound site was dominated by fibroblasts, with the presence of dense type I collagen. This indicated that these groups were in transition from late proliferation to early remodeling. In addition, the decrease in the number of blood vessels after 2 weeks of healing occurred to facilitate the reorganization of the collagen matrix [42].

## 5. Conclusions

In this study, new sucrose-impregnated G/M films were prepared and evaluated as promising dressings that can improve the healing performance of deep wounds. These sucrose-impregnated G/M films significantly improved the maintenance of hydration, G/M crystallinity, fracture stress, and elasticity due to stronger intermolecular attraction between polymer sheets. In addition, the sucrose-impregnated G/M dressing also had WVTR and surface roughness comparable to G/M-H_2_O dressings, which is optimal for effective wound healing. In in vivo studies, sucrose-impregnated G/M dressings showed significantly improved healing performance by promoting earlier initiation of re-epithelialization, deposition, and maturation of collagen fibers than control G/M-H_2_O dressings. The wound healing of G/M-70% sucrose was significantly more vascular, and matured early. The healing of the wound with G/M-70% sucrose applied was evidenced by significantly more blood vessels, an early mature regenerating epidermis, and a higher density of oriented collagen fibers. Thus, these results demonstrate that G/M-70% sucrose may be a promising candidate for deep wound healing.

## Figures and Tables

**Figure 1 bioengineering-12-00327-f001:**
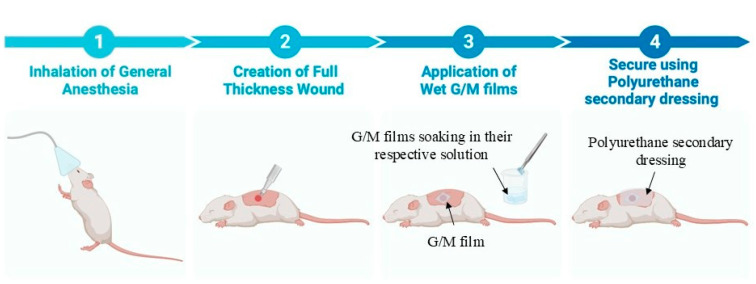
Schematic illustration of procedure used in animal surgery.

**Figure 2 bioengineering-12-00327-f002:**
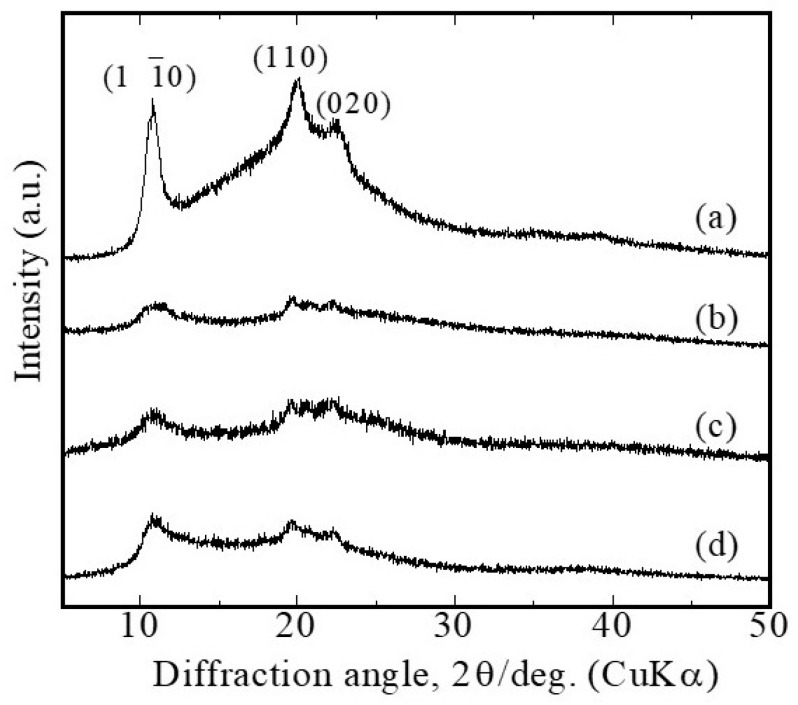
XRD patterns of G/M in (**a**) dry state, (**b**) hydrated state in distilled water, (**c**) 30% sucrose solution, and (**d**) 70% sucrose solution.

**Figure 3 bioengineering-12-00327-f003:**
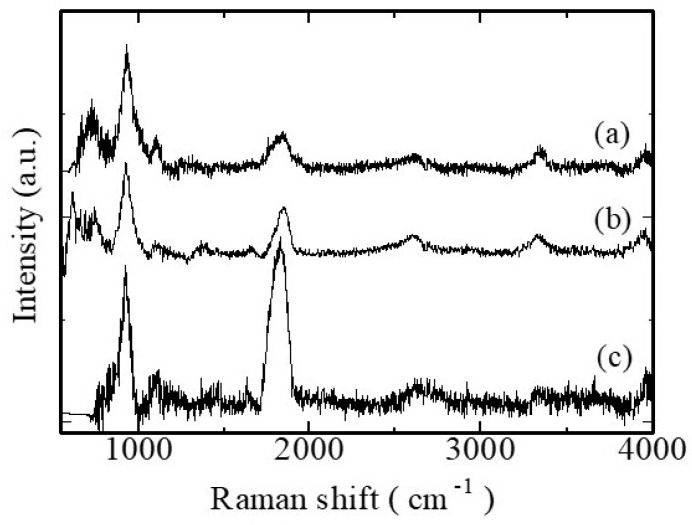
Raman spectra of (**a**) dry G/M and dehydrated G/M after soaking in: (**b**) 30% sucrose and (**c**) 70% sucrose solutions.

**Figure 4 bioengineering-12-00327-f004:**
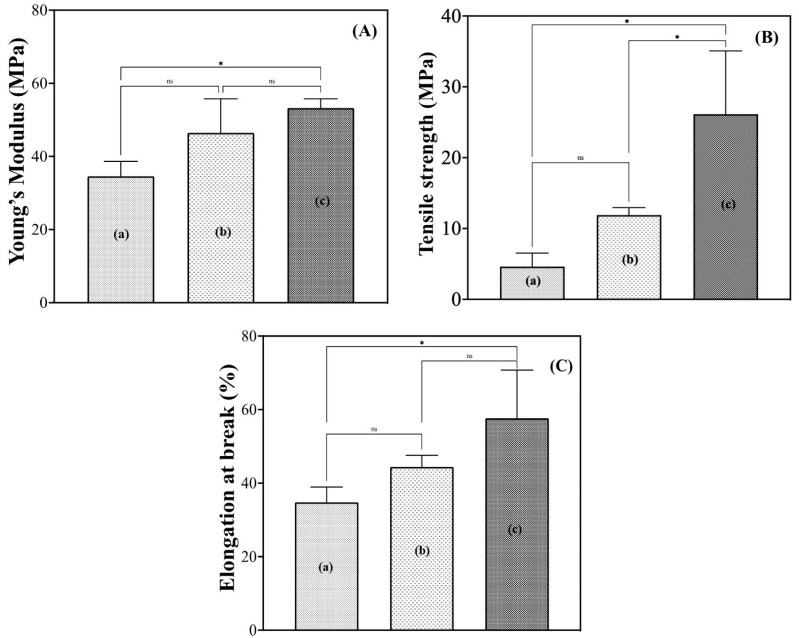
(**A**) Young’s modulus, (**B**) tensile strength, and (**C**) elongation of G/M hydrated in (a) distilled water, (b) 30% sucrose, and (c) 70% sucrose. *: *p* < 0.05, ns: not significant.

**Figure 5 bioengineering-12-00327-f005:**
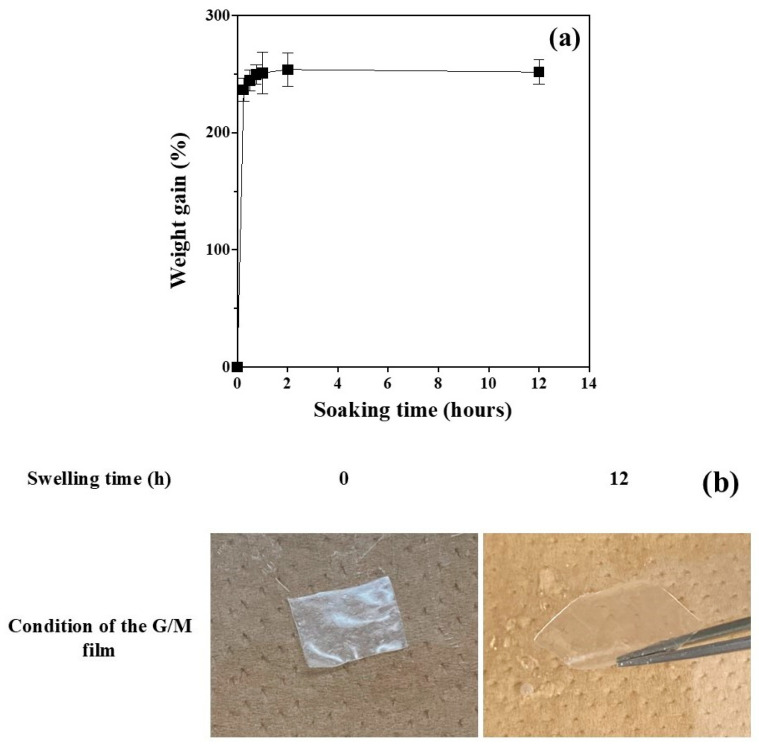
(**a**) Weight gain of G/M film with soaking time, and (**b**) photographs of G/M film before and after swelling in phosphate-buffered saline for 12 h.

**Figure 6 bioengineering-12-00327-f006:**
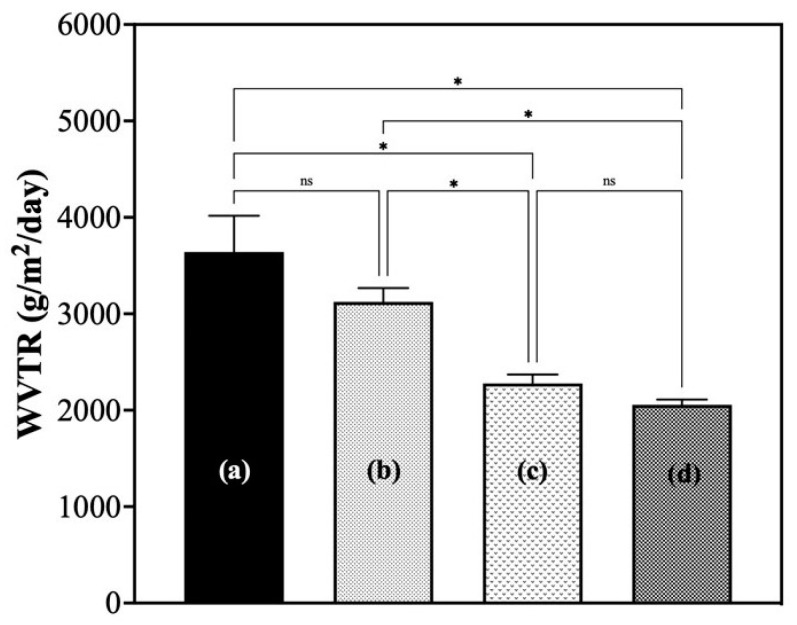
Water vapor transmission rate of (a) control, (b) G/M-H_2_O, (c) G/M-30% sucrose, and (d) G/M-70% sucrose at 37 °C for 14 days. All groups except control group consist of polyurethane secondary dressing. *: *p* < 0.05, ns: not significant.

**Figure 7 bioengineering-12-00327-f007:**
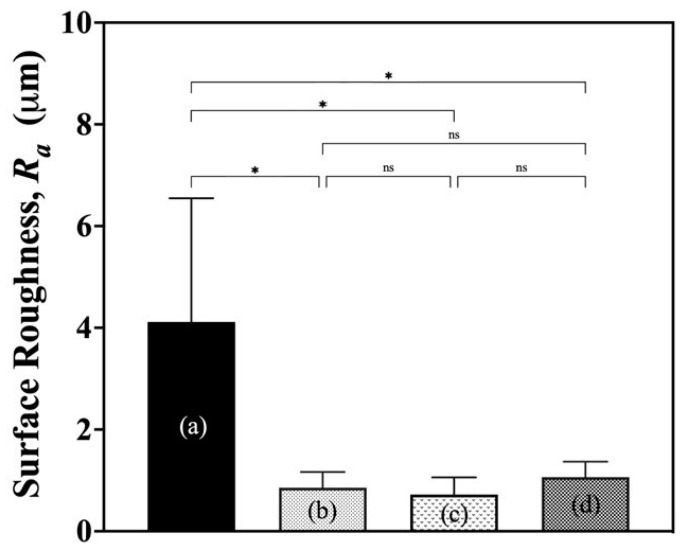
Surface roughness of G/M in (a) dry state and after soaking in (b) distilled water, (c) 30% sucrose, and (d) 70% sucrose. *: *p* < 0.05, ns: not significant.

**Figure 8 bioengineering-12-00327-f008:**
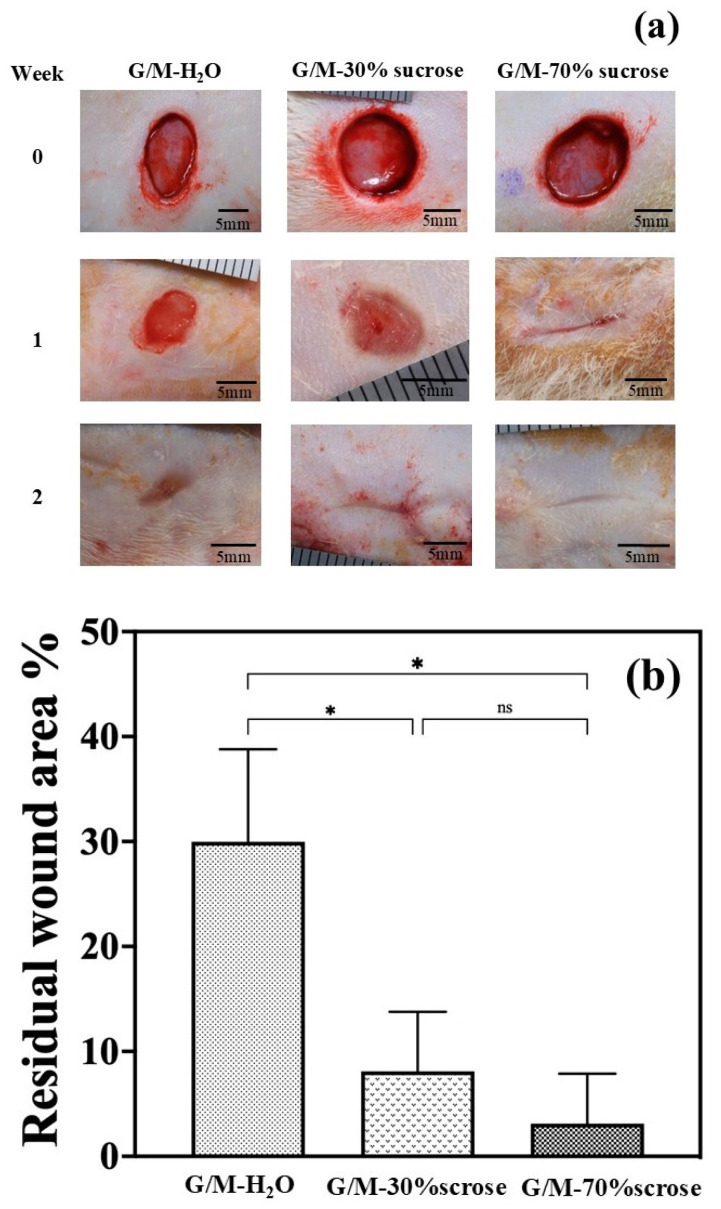
(**a**) Macroscopic observation of wound in each sample group at 0, 1, and 2 weeks post-operation. (**b**) Residual wound area of each sample group at 1 week post-operation. *: *p* < 0.05, ns: not significant.

**Figure 9 bioengineering-12-00327-f009:**
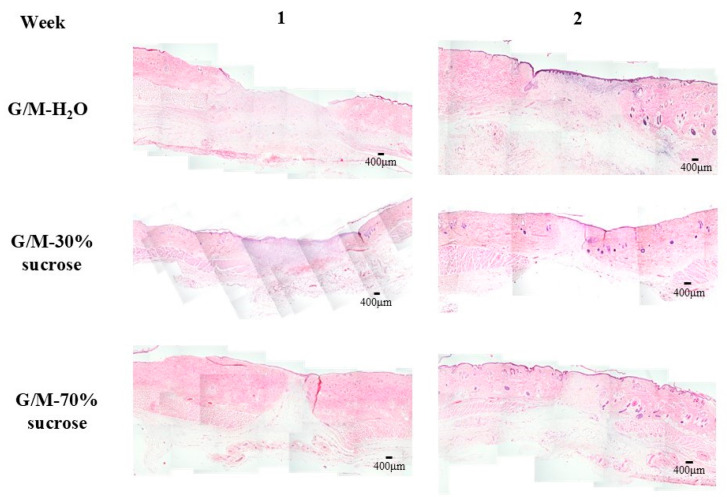
HE staining images of regenerated and surrounding tissue.

**Figure 10 bioengineering-12-00327-f010:**
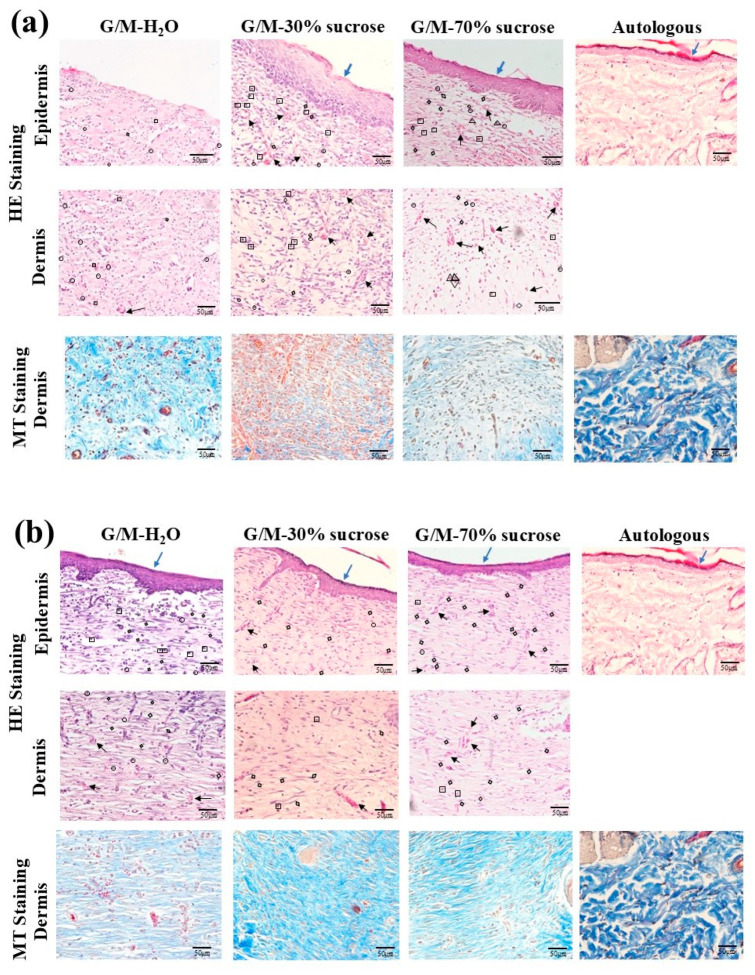
HE and MT staining images of the regenerated tissue at (**a**) 1 week and (**b**) 2 weeks. The symbols of △, ○, □, and ◇ represent lymphocytes, neutrophils, macrophages, fibroblasts, respectively. Arrows↑ indicate blood vessels.

**Figure 11 bioengineering-12-00327-f011:**
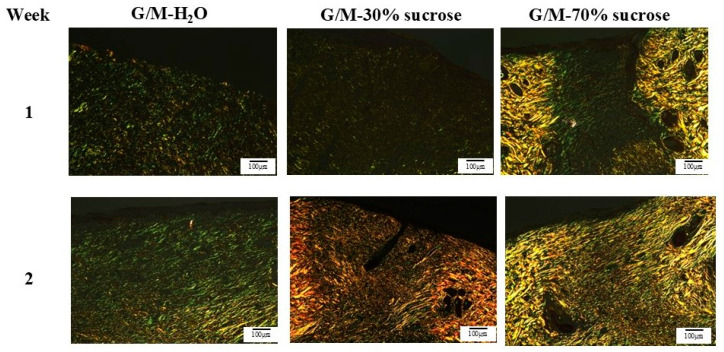
PSR staining images of regenerated tissue at 1 and 2 weeks. The green and yellow areas in this stained image represent type III collagen and type I collagen, respectively.

**Figure 12 bioengineering-12-00327-f012:**
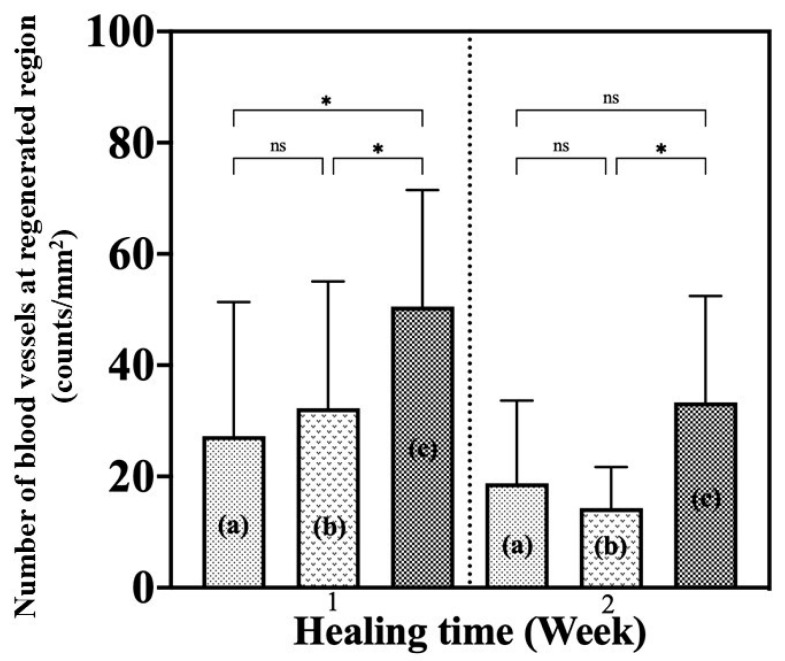
Number of blood vessels in the regenerated tissue region at 1 and 2 weeks post-operation: (a) G/M-H_2_O, (b) G/M-30% sucrose, and (c) G/M-70% sucrose. *: *p* < 0.05, ns: not significant.

**Table 1 bioengineering-12-00327-t001:** Composition of sucrose solutions at different concentrations.

Sucrose Solution Concentration % (*w*/*w*)	Weight of Sucrose Granules (g)	Volume of Distilled Water (mL)
30	3	10
70	7	10

**Table 2 bioengineering-12-00327-t002:** List of abbreviations of G/M samples and descriptions of soaking conditions.

Abbreviation of Wound Dressing Group	Description
G/M	G/M in dry condition
G/M-H_2_O	G/M immersed in water
G/M-30% sucrose	G/M immersed in 30% sucrose solution
G/M-70% sucrose	G/M immersed in 70% sucrose solution

## Data Availability

The original contributions presented in this study are included in the article. Further inquiries can be directed to the corresponding author.

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
