# Peer review of "Characterization of Sucrose-Impregnated Crystalline Glucose/Mannose Films as Moisturizing Wound Dressings and Their Significant Healing Effect on Deep Wounds in a Rat Model"

_bioengineering, 2025, doi:10.3390/bioengineering12040327_

Round 1
Reviewer 1 Report
Comments and Suggestions for Authors
This study introduces sucrose-impregnated glucose/mannose (G/M) films to enhance deep wound healing. Sucrose improved crystallinity, mechanical strength, and healing performance, with the 70% sucrose formulation showing the best results in a rat model. The findings suggest these films as promising wound dressings. This is an interesting and well-written article, but it requires major revisions in certain sections.
Introduction:
- While you mention their moisture retention properties and ECM similarity, additional details on their specific role in wound healing (e.g., antimicrobial properties or specific interactions with skin cells) could strengthen the argument.
- You describe its water activity effects and role in attracting macrophages and fibroblasts, but do not elaborate on how these translate into enhanced healing outcomes.
- The introduction states that no reports exist on the combination of glucose/mannose and sucrose, but you could clarify why combining them is hypothesized to be beneficial.
Study design:
- The study uses six rats divided into three groups, meaning each group has only two rats. This is a very small sample size, which may limit statistical power and generalizability. Consider increasing the number of animals per group to improve reliability.
- It would be beneficial to include a **negative control** (e.g., untreated wounds or wounds treated with a standard dressing) to provide a baseline comparison for the healing effects of the G/M films.
- The study assesses healing at weeks 1 and 2, but longer-term evaluation (e.g., week 4) might be useful to determine full tissue regeneration.
- Why were 30% and 70% sucrose chosen? Providing a rationale for these specific concentrations based on prior literature or preliminary studies would strengthen the study's foundation.
- While histological evaluation helps assess healing, additional markers of inflammation or immune response (e.g., cytokine analysis) could enhance the biocompatibility assessment.
Material and methods:
- The sterilization process (UV sterilization) is mentioned but not clearly detailed regarding its duration and frequency. Consider specifying these aspects.
- The drying condition after sterilization (60°C) is provided, but whether this is under vacuum or open-air conditions is unclear.
- The mechanical strength testing provides clear measurement conditions, but it would help to mention how many samples were tested per group for statistical reliability.
- The number of animals used (six) is relatively small. Were any sample size calculations performed to determine statistical power?
- The **macroscopic evaluation** describes how images are analyzed, but were evaluators blinded to the treatment groups?
- The **histological analysis** method is clear, but additional information on image analysis (e.g., software or criteria for measuring collagen deposition) would improve clarity.
- The statistical methods are appropriate, but it would be helpful to clarify whether normality checks were conducted before using ANOVA.
Results:
- While you present numerical data well, ensuring that each data point is directly tied to its biological significance would strengthen the interpretation. For example, explicitly stating how the mechanical and absorption properties contribute to wound healing effectiveness.
- Make sure that all figures are referenced in a logical order and that their descriptions clearly highlight the key takeaways. If a reader only looked at the text, would they fully grasp the significance of the results?
Discussion and conclusion:
The results effectively back up the discussion and conclusion. They clearly demonstrate the structural, mechanical, and biological benefits of sucrose-impregnated G/M films, providing a strong foundation for the claim that these films significantly improve wound healing. If there is any concern, it may be in ensuring that limitations (e.g., potential long-term effects, comparisons with existing treatments) are acknowledged in the discussion.
Comments on the Quality of English LanguageThe quality of English in this manuscript is generally good, but there are some areas where improvements can be made for clarity, conciseness, and grammatical correctness.
- Some sentences are too long and could benefit from being broken into shorter, more digestible segments.
_"Crystalline glucose/mannose film (G/M) had demonstrated exclusive water absorptivity and vapor transmission rate, with remarkable wound healing performance."_
- Certain phrases are redundant or awkwardly worded.
_"The degree of crystallinity of hydrated G/M increases with increasing sucrose concentration."_
_"Besides, the in-vivo study demonstrated healing performance in the following order: G/M-70% sucrose > G/M-30% sucrose > G/M-H2O."_
- The manuscript shifts between past and present tense inconsistently.
_"This study evaluated the additive effect of sucrose on G/M at different concentrations..."_
If you are reporting completed research, past tense is correct. However, in general descriptions and interpretations, present tense may be preferable.
- Grammar & Syntax Issues:
- "The diffraction patterns of G/M in all conditions showed three peaks at 2θ approximately 10°, 20° and 22°."
- "This phenomenon corresponded to the increase of the interlayer distance."
- "This was attributed to the decrease in crystallinity, which is largely due to the infil- tration of water and sucrose molecules into the crystalline layer, interacting with the hy- droxyl groups in the backbone of G/M."
- Word Choice & Clarity:
- "Furthermore, it can be observed that the intensity of the peaks decreased with the peak broadening that occurred when G/M were in hydrated states."
- "Besides, the decrease in surface roughness of hydrated G/M, makes it a suitable candidate as wound dressing, as it will not cause any irritation and inflammation when apply to the wound."
- Tense Consistency:
- "The intensity of these peaks also increases for G/M-70% sucrose group."
- Redundancies & Repetitions:
- "The surface roughness of G/M films in dry and hydrated states were determined as shown in Fig. 7."
(This sentence is repeated twice in the paragraph; one should be removed.)
Author Response
Dear, Reviewer
Thank you for your valuable comments. We responded to all reviewer comments and the parts that were revised based on comments from the reviewer are shown in red. We are confident that the reviewer's valuable comments made this article even better. Please see the file. Detailed responses to the reviewer can be found in this file.

Reviewer 2 Report
Comments and Suggestions for Authors
Dear Authors,
The present research paper entitled “Characterization of sucrose-impregnated crystalline glucose/mannose films and their significant healing effect on deep wounds in a rat model” is an in vivo study that included materials characterization experimental work, which investigates the effect of sucrose incorporation into crystalline glucose/mannose wound dressing material’s mechanical, chemical, crystal structure and healing features. The findings in the present work are interesting and were described clearly for the readers. Thus, the present form of the manuscript can be published in your prestigious journal after minor corrections. The recommendations can be seen below;
- On page 2 of 20, in the Introduction section on the second paragraph, the authors should also add literature to discuss the mechanical performance of these materials, which makes them promising materials for wound healing dressings.
- On page 3 of 20, in section 2.4, this part is not clear. Is there any mixing process to combine both solutions for 30 mins, or was it immersed in a solution during the 30 minutes?
- On page 5 of 20, what is the secondary dressing used in the present work, which needs to be identified in the text?
- On page 5 of 20, in 2.5.6 surface roughness, this part needs to be detailed more.
- On page 7 of 20, Results section should also include film preparation process details.
- On page 11 of 20, last paragraph needs to be discarded due to a repeat form of the previous paragraph.
- On page 11 of 20, in the first sentence of the last paragraph, the authors should remove these references because it confuses the readers to understand whether Figure 11 shows the results from the literature or they are the results achieved in this experimental work.
Kind regards,
Author Response
Dear Reviewer
Thank you for your valuable comments. We responded to all reviewer comments and the parts that were revised based on comments from the reviewer are shown in red. Please see the file. Detailed responses can be found in the file.

Round 2
Reviewer 1 Report
Comments and Suggestions for Authors
The authors have addressed all comments correctly. The article is now suitable for printing in its current form.